# Investigating the causal effects of COVID-19 vaccination on the adoption of protective behaviors in Japan: Insights from a fuzzy regression discontinuity design

**Fengming Chen**[1]*, **Hayato Nakanishi**[2], **Yoichi Sekizawa**[3], **Sae Ochi**[4], **Mirai So**[5]

**1** Graduate School of Economics and Management, Tohoku University, Aoba-ku Kawauchi, Sendai-shi, Miyagi, Japan, **2** Faculty of Economics, Kanagawa University, Rokkakubashi Kanagawa-ku, Yokohama, Japan, **3** Research Institute of Economy, Trade and Industry, Chiyoda-ku, Kasumigaseki, Tokyo, Japan, **4** Department of Laboratory Medicine, The Jikei University School of Medicine, Minato-ku Nishishinbashi, Tokyo, Japan, **5** Department of Psychiatry, Tokyo Dental College Ichikawa General Hospital, Ichikawa-shi, Sugano, Chiba, Japan

* fengming.chen.d2@tohoku.ac.jp

**Data Availability Statement:** The data used in the present study belong to the RIETI and can be

## Abstract

### Background

During the COVID-19 pandemic, concerns emerged that vaccinated individuals might engage less in infection-preventive behaviors, potentially contributing to virus transmission. This study evaluates the causal effects of COVID-19 vaccination on such behaviors within Japan, highlighting the significance of understanding behavioral dynamics in public health strategies.

### Methods

Utilizing Japan's age-based vaccination priority for those born before April 1, 1957, this research employs a regression discontinuity design (RDD) to assess the vaccination's impact. Data from the fourth round of a longitudinal online survey, conducted from July 20 to 27, 2021, served as the basis for analyzing 14 infection-protective behaviors, including mask usage, handwashing, and avoiding crowds.

### Results

A total of 12067 participants completed the survey. The analyzed sample size varied by outcome variable, ranging from 1499 to 5233. The analysis revealed no significant differences in the 14 behaviors examined among fully vaccinated, partially vaccinated, and unvaccinated individuals. This consistency across groups suggests that vaccination status did not significantly alter engagement in protective behaviors during the observation period.

### Conclusions

Empirical findings highlight the complexity of behavioral responses following vaccination, indicating that such responses may be influenced by various factors, rather than by

obtained from the institute upon reasonable request (info@rieti.go.jp).

**Funding:** This work was supported JST Grant Number JPMJPF2201 to CHEN. The funder had no role in study design, data collection and analysis, decision to publish, or preparation of the manuscript.

**Competing interests:** The authors have declared that no competing interests exist.

vaccination status alone. Additionally, this result underscores the importance of crafting public health policies that account for the intricate interplay between vaccination and behavior. This study contributes to the broader discourse on managing responses to the pandemic and tailoring interventions to sustain or enhance protective health behaviors amid vaccination rollouts.

## Introduction

The emergence of COVID-19 toward the end of 2019 led to significant loss of life and profound alterations in behavior and daily routines. Numerous nations enforced stringent measures to curb the disease's transmission, including lockdowns and the mandatory use of face masks. Even in countries where strict government measures were not in place, many people voluntarily adopted protective behaviors, such as the widespread use of face masks in Japan.

While vaccination decreases the likelihood of hospitalization and may mitigate COVID-19 symptoms, it is not an infallible solution to the issue of COVID-19. This is because, following vaccination, some individuals continue to exhibit weak immune responses [1]. Hence, there is a concern that individuals who believe vaccination will lower their risk of infection or complications may take fewer protective measures, potentially resulting in increased virus transmission [2,3]. Engagement in such potentially unsafe behaviors is known as the Peltzman effect [4] or risk compensation [5].

Previous findings on risk-compensation behaviors due to COVID-19 vaccination have been inconsistent. One study showed a positive association between vaccine coverage and mobility using a longitudinal dataset of 107 countries [6]. Similarly, other researchers used a case-crossover study to report that non-household contact and non-essential shopping increased after vaccination in England and Wales [7]. Moreover, one study from China reported that the protective behaviors of Chinese university students weakened after vaccination [8]. In contrast, using Japan's longitudinal data, Yamamura et al. [9] reported that vaccinating led to staying home more but did not change handwashing and mask-wearing practices. Previous findings further indicated that vaccinated Canadian individuals were more likely to report engaging in distancing, wearing masks, and maintaining hand hygiene compared with unvaccinated individuals [10]. However, other studies found no association between vaccination and protective behaviors in the UK [11,12], China [13], Thailand [14], and the US [15] or in a cross-section study of 12 countries [16]. All these previous studies used observational data for their analyses.

However, estimating the effect of vaccination on people's protective behaviors against COVID-19 using observational data is difficult because of the inherent differences between vaccinated and unvaccinated individuals. To address this difficulty, we applied a regression discontinuity design (RDD), a quasi-experimental design exploring causal effects from observational data [17]. RDD is employed in cases where an intervention is conducted only when the participants' measure of some variable (running variable) exceeds a specific cutoff value. The RDD evaluates the impacts of interventions on outcome variables in the vicinity of the cutoff.

In Japan, those born on April 1, 1957, or before (approximately aged ≥65 years) were given priority in receiving vaccinations against COVID-19. Therefore, in the early stage of vaccine rollout, the ratio of those vaccinated was higher among individuals born in March 1957 or before than among those born after that month. Consequently, we applied the RDD using birth month as the running variable with a cutoff of March 1957. Several studies have

estimated the causal relationships between influenza, COVID-19 vaccinations, and health outcomes using RDD with age or birth timing as the running variables [18–21]. However, to the best of our knowledge, few studies have investigated the causal relationship between COVID-19 vaccination and protective behaviors against COVID-19.

Therefore, in this study, we applied the RDD to investigate whether vaccination against COVID-19 affects people's protective behaviors against COVID-19, particularly in wearing facemasks, handwashing, and avoiding going outside.

## Methods

### Japan's vaccination procedure

The Japanese government decided that the first COVID-19 vaccine should be administered sequentially in the following order: vaccination opportunities were provided first to healthcare workers; second to those born before April 1, 1957; and third to those with pre-existing medical conditions or working in facilities for older adults. COVID-19 vaccination for medical personnel in Japan commenced on February 17, 2021, and began on April 12 for individuals born on or before April 1, 1957. The local governments and Self-Defense Forces provided vaccinations in line with this rule. In addition, workplace vaccination opportunities were provided, particularly to employees of large firms or organizations and their families, regardless of age [22].

This decision by the government based on birth created two groups that faced different opportunities to receive vaccines, even with a minimal time difference between their births. Specifically, those born before April 1, 1957, reaching 65 years old by the end of March 31, 2022, were prioritized for vaccination compared with those born after April 2, 1957, unless they were healthcare workers. As vaccination was not mandatory, some people older than 65 years did not receive vaccination, whereas some people younger than 65 years received vaccination at an early stage, such as healthcare workers and those who could receive workplace vaccination. Therefore, it was hypothesized that individuals born in March 1957 or before were more likely to have been vaccinated against COVID-19 than those born in April 1957 or after in the early stages of vaccine provision. As we could not confirm this hypothesis based on publicized data, we examined its validity using our original data, as mentioned below.

### Study data

As an integral component of the research initiative led by the Research Institute of Economy, Trade, and Industry in Japan (RIETI), some of the authors (mainly YS who was employed by RIETI) carried out the longitudinal online survey titled "Continuing Survey on Mental and Physical Health during the COVID-19 Pandemic" (hereinafter, "RIETI questionnaire survey"). NTTCom Online Marketing Solutions Corporation carried out this survey following the design and instructions provided by RIETI mainly through YS. The survey was conducted in five rounds: Round 1 from October 27 to November 6, 2020; Round 2 from January 19 to 26, 2021; Round 3 from April 23 to May 6, 2021; Round 4 from July 20 to 27, 2021; and Round 5 from October 20 to 27, 2021. Participants for the study were selected from individuals registered with NTTCom Online Marketing Solutions Corporation or its affiliates. After completing the survey, the authors, including YS, applied to RIETI for permission to use the dataset, as it belonged to RIETI. Following an internal review, RIETI granted the authors access to the dataset.

Previous studies comprehensively describe the data collection method employed [23,24]. During the first round of the survey, participants were screened as individuals aged 18–74 who were currently residing in Japan. They were chosen in a manner that ensured that their demographic composition ratios regarding gender, age, and distribution of residential prefectures

closely mirrored the population estimates presented by the Statistics Bureau of Japan (final estimates, May 2020); 16,642 individuals (8,022 men and 8,620 women) were established as valid respondents. NTTCom Online Marketing Solutions Corporation requested that all eligible respondents from the first round of the survey participate in the subsequent survey rounds via email based on the instruction from RIETI. All interactions with the study participants, including the questionnaires, were conducted in Japanese. This study used the first and fourth rounds of the RIETI questionnaire survey.

Every individual involved in this survey granted written informed consent online for their participation. The ethics committee of Hiramatsu Memorial Hospital approved this survey (no. 20200925). The present study was also approved by the ethics committee of Hiramatsu Memorial Hospital(no.20230626).

The dataset utilized in our study, while under the stewardship of RIETI, is not openly accessible because of specific constraints. Despite our involvement in the survey process, we also had to apply for access to the data. RIETI has not sanctioned the data's public release for two primary reasons. First, RIETI lacks established policies or guidelines for public data dissemination. Second, there is a risk of contravening the Japanese Personal Information Protection Law and the agreement with the study participants, as the dataset, although anonymized, contains variables (e.g., birth year and month, residence postal code) that could reveal personal identities. Nevertheless, RIETI permits data access to researchers who comply with strict confidentiality management criteria and adhere to procedural requirements[25,26]. Researchers interested in accessing the data can contact RIETI via email (info@rieti.go.jp).

## Vaccination variables

The vaccination-related variables used in this study are derived from the following two questions in Round 4 of the RIETI questionnaire survey. The first question inquired whether the participants had received prior vaccination. Participants were categorized as vaccinated if they had received a minimum of one dose. The second question applied only to participants who had been vaccinated and asked how many times they had received a vaccine against COVID-19. Using these questions, we generated a binary variable with a value of 1 for respondents who had received at least two vaccinations and 0 for respondents who had received either no doses or just one dose of the COVID-19 vaccine.

## Outcome variables

The outcome variables were 12 dichotomous variables and two ordered categorical variables created from questions on potentially protective behaviors. The dichotomous variables were created from the answers regarding 12 potentially protective behaviors against COVID-19 infection: avoiding going to poorly ventilated places, avoiding going to crowded places, avoiding conversing or vocalizing near others, wearing a mask, handwashing, sanitizing hands, changing clothes frequently, gargling, sanitizing personal belongings, keeping people at a distance when going out, refraining from visiting medical facilities, and avoiding going outside. The participants were asked to check whether their answer was yes for each of the 12 items. We generated a dichotomous variable with 1 for those who selected yes for an item and 0 otherwise. Two ordered categorical variables were created using the following questions: The RIETI questionnaire survey asked another question regarding going out: "How often have you gone out in the past month?" The answers were "1. Almost every day," "2. 4–5 days per week," "3. 2–3 days per week," "4. 1 day per week," "5. 1 day per month," and "6. Not at all." For the analyses, we reversed the order ("almost every day" as six and "not at all" as one) and treated the variable as a continuous one. Similarly, we asked, "How frequently have you been directly

meeting acquaintances, such as relatives, friends, and neighbors, excluding those living with you, in the past month, apart from work-related interactions?" The answers ranged from "1. Almost every day" to "6. Not at all." We reversed the order here as well and treated this variable as continuous.

Of these variables, the dichotomous variables of wearing a mask, handwashing, and avoiding going out were treated as the primary outcomes. The other outcome variables were secondary outcomes. Details of these questions are presented in the S1 Table. All of these outcome measures were from the fourth round of the survey.

## Statistical analyses

We estimated the effects of being vaccinated twice on potential protective behaviors against COVID-19 by applying the RDD, using participants' birth month as the running variable and a cutoff of March 1957. To be more precise, our analysis was a fuzzy RDD because surpassing the cutoff does not result in a perfect shift from no vaccination to vaccination, as mentioned above.

RDD analyses are feasible when the running variable is not manipulated near the cutoff [27]. To confirm that there was no evidence of manipulation, we conducted visual checks using histograms around the cutoff as well as the statistical manipulation tests [28].

Upon confirming that there was no manipulation near the cutoff, we examined the validity of the hypothesis that individuals born in March 1957 or before were more likely to have been vaccinated against COVID-19 than those born in April 1957 or after in the early stages of vaccine provision. We applied a local linear regression, a triangular kernel function, which assigned varying weights to data points based on their proximity to the cutoff point and selected the mean square error optimal bandwidth [29].

Upon confirming that there was a significant jump in the vaccination rate for the birth month of March 1957, we estimated the effect of being vaccinated twice on the outcome variables by applying a fuzzy RDD using birth in March 1957 or before as an instrumental variable. Using fuzzy RDD, we obtained the local average treatment effect (LATE) on "compliers," who would not be treated if their values were under the cutoff and would be treated if their values were beyond the cutoff.

Then, the following secondary analyses were conducted. First, we created another variable: those vaccinated once or more with the value 1 and those not vaccinated at all with the value 0. Second, we conducted a placebo test using the outcome variables from the first round of the survey.

Covariate adjustments were not incorporated into our analyses, as they are unnecessary for RDD [27,30]. We provided 95% confidence intervals (CI) and *p*-values utilizing the robust bias-corrected standard errors [27]. All statistical analyses were executed using the "rddensity," "rdrobust," and "rdplot" commands [28,30] within Stata 17 (Stata Corp, College Station, TX, USA). The level of statistical significance was defined as $p < 0.05$.

## Results

### Characteristics of study participants

The characteristics of the study participants in the first and fourth survey rounds are listed in Tables 1 and S2, respectively. The fourth round of the survey was completed by 12067 participants. However, the analyzed sample size differed across outcome variables, spanning from 1499 to 5233.

**Table 1. Characteristics of study participants in the fourth round of the survey.**

| | | Born from April 1957 to March 1962 (n = 1,603) | Born from April 1962 to March 1967 (n = 1,365) | Total (N = 12,067) |
|---|---|---|---|---|
| **Age, years** | | 61.7 (1.4) | 66.6 (1.5) | 53.6 (14.1) |
| **Gender** | Men | 848 (52.9%) | 715 (52.4%) | 6,186 (51.3%) |
| | Women | 755 (47.1%) | 650 (47.6%) | 5,881 (48.7%) |
| **Highest education level, No. (%)** | Junior/senior high school | 435 (27.1%) | 420 (30.8%) | 3,728 (30.9%) |
| | Two- or three-year college | 361 (22.5%) | 277 (20.3%) | 2,594 (21.5%) |
| | Four-year college or higher | 807 (50.3%) | 668 (48.9%) | 5,745 (47.6%) |
| **Marital status, No. (%)** | Married | 1,178 (73.5%) | 1,053 (77.1%) | 7,617 (63.1%) |
| | Divorced | 131 (8.2%) | 102 (7.5%) | 721 (6.0%) |
| | Bereaved | 50 (3.1%) | 66 (4.8%) | 305 (2.5%) |
| | Never married | 244 (15.2%) | 144 (10.5%) | 3,424 (28.4%) |
| **Employed, %** | | 62.3% | 42.9% | 60.9% |
| **Lifestyle, %** | Avoiding going to poorly ventilated places | 90.1% | 93.3% | 88.9% |
| | Avoiding going to crowded places | 91.1% | 92.7% | 90.1% |
| | Avoiding conversing or vocalizing near others | 88.5% | 89.7% | 86.7% |
| | Wearing a mask | 98.6% | 98.3% | 97.5% |
| | Handwashing | 96.1% | 97.6% | 95.9% |
| | Sanitizing hands | 90.8% | 91.4% | 90.9% |
| | Changing clothes frequently | 26.4% | 25.8% | 28.2% |
| | Gargling | 65.9% | 69.5% | 69.0% |
| | Sanitizing personal belongings | 27.3% | 26.7% | 33.5% |
| | Keeping people at a distance when going out | 87.5% | 89.7% | 85.7% |
| | Refraining from visiting medical facilities | 43.7% | 44.4% | 47.9% |
| | Avoiding going outside | 66.1% | 70.8% | 67.7% |
| **Frequency of going out, No. (%)** | Almost everyday | 492 (30.7%) | 328 (24.0%) | 3,458 (28.7%) |
| | 4–5 days per week | 412 (25.7%) | 373 (27.3%) | 3,185 (26.4%) |
| | 2–3 days per week | 395 (24.6%) | 394 (28.9%) | 2,881 (23.9%) |
| | 1 day per week | 233 (14.5%) | 203 (14.9%) | 1,755 (14.5%) |
| | 1 day per month | 30 (1.9%) | 37 (2.7%) | 369 (3.1%) |
| | Not at all | 41 (2.6%) | 30 (2.2%) | 419 (3.5%) |
| **Frequency of meeting acquaintances, No. (%)** | Almost everyday | 80 (5.0%) | 74 (5.4%) | 738 (6.1%) |
| | A few times per week | 241 (15.0%) | 260 (19.0%) | 1,792 (14.9%) |
| | Once per week | 202 (12.6%) | 250 (18.3%) | 1,535 (12.7%) |
| | Once per two weeks | 171 (10.7%) | 137 (10.0%) | 1,174 (9.7%) |
| | Once per month | 280 (17.5%) | 201 (14.7%) | 1,992 (16.5%) |
| | Not at all | 629 (39.2%) | 443 (32.5%) | 4,836 (40.1%) |
| **Frequency of vaccination, No. (%)** | 0 | 827 (51.6%) | 224 (16.4%) | 6,768 (56.1%) |
| | 1 | 597 (37.2%) | 314 (23.0%) | 2,413 (20.0%) |
| | 2 | 179 (11.2%) | 827 (60.6%) | 2,886 (23.9%) |

## Confirmation of the validity of applying fuzzy RDD

After conducting visual examinations of the histograms, no indications of manipulation around the cutoff were discovered (S1 Fig). Based on the manipulation test [28], the *p*-value

calculated from the null hypothesis and test statistics was 0.537. Thus, the null hypothesis that there was no manipulation was not rejected at the 5% significance level.

Fig 1 presents the RD estimate of the impact of being born in March 1957 or before on the probability of receiving the vaccine twice in the fourth round of the survey. There was a dramatic jump in the probability at the cutoff. The RDD analysis shows that the gap is 47.3% points (95% CI: 39.4 to 57.2) (S3 Table). Thus, we can utilize the gap observed in the fourth survey round around the cutoff to identify the effects of vaccination on respondents' protective behaviors against COVID-19.

## Results on primary outcomes

Table 2 presents the effects of being vaccinated twice on primary outcome variables using fuzzy RDD analyses. There were no significant differences between those being vaccinated twice and those not: wearing a mask (0.00, 95% CI: -0.06 to 0.08), handwashing (0.03, 95% CI: -0.05 to 0.12), and avoiding going outside (0.06, 95% CI: -0.13 to 0.26). Results on other outcome variables are also presented in Table 2. These results were consistent with the primary outcomes. None of the protective behaviors were significantly influenced by vaccination.

## Results of secondary analyses

First, we treated individuals vaccinated at least once as the intervention group; otherwise, the individuals belonged to the control group. S4 Table shows a noticeable jump in the vaccination rates around the threshold, similar to the previous analysis. No significant results were observed for outcome variables (S5 Table).

Second, we conducted a placebo test using the protective behaviors in the first survey round as outcome variables. The estimation results show that receiving vaccination twice was not

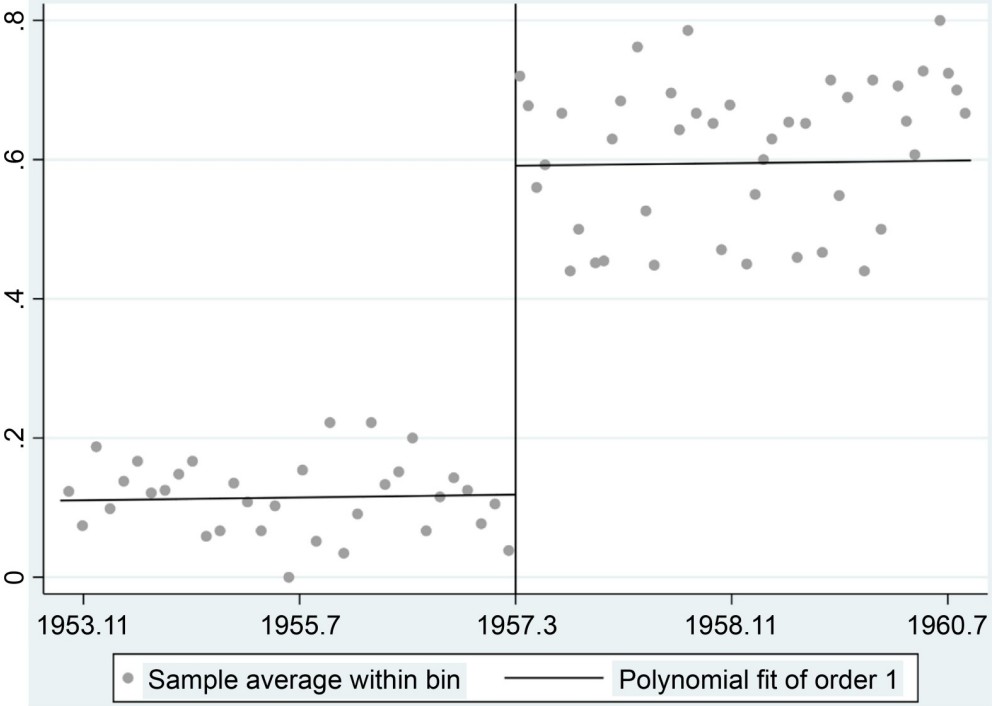

**Fig 1. RD plot of the probability of receiving the vaccine twice.**

**Table 2. Estimation results of the regression discontinuity design on outcomes.**

| Outcome variable | Point estimate | 95% CI | *p*-value | Bandwidth (months) | Total sample | Analyzed samples | |
|---|---|---|---|---|---|---|---|
| | | | | | | Control | Treatment |
| **Wearing a mask** | 0.00 | (-0.06–0.08) | 0.757 | 57.70 | 12,067 | 1,527 | 1,302 |
| **Handwashing** | 0.03 | (-0.05–0.12) | 0.394 | 55.55 | 12,067 | 1,483 | 1,269 |
| **Avoiding going outside** | 0.06 | (-0.13–0.26) | 0.490 | 55.19 | 12,067 | 1,483 | 1,269 |
| **Avoiding going to poorly ventilated places** | 0.04 | (-0.09–0.16) | 0.539 | 54.75 | 12,067 | 1,465 | 1,242 |
| **Avoiding going to crowded places** | -0.05 | (-0.17–0.05) | 0.318 | 66.55 | 12,067 | 1,753 | 1,515 |
| **Avoiding conversing or vocalizing near others** | 0.04 | (-0.10–0.20) | 0.529 | 48.30 | 12,067 | 1,325 | 1,121 |
| **Sanitizing hands** | 0.04 | (-0.07–0.17) | 0.392 | 64.39 | 12,067 | 1,705 | 1,472 |
| **Changing clothes frequently** | -0.14 | (-0.39–0.02) | 0.073 | 39.78 | 12,067 | 1,091 | 954 |
| **Gargling** | -0.03 | (-0.25–0.14) | 0.568 | 59.37 | 12,067 | 1,583 | 1,355 |
| **Sanitizing personal belongings** | 0.03 | (-0.14–0.18) | 0.808 | 75.55 | 12,067 | 1,954 | 1,702 |
| **Keeping people at a distance when going out** | 0.01 | (-0.12–0.14) | 0.858 | 63.05 | 12,067 | 1,678 | 1,447 |
| **Refraining from visiting medical facilities** | 0.08 | (-0.12–0.25) | 0.491 | 73.10 | 12,067 | 1,904 | 1,650 |
| **Frequency of going out** | -0.47 | (-1.06–0.06) | 0.082 | 46.93 | 12,067 | 1,287 | 1,084 |
| **Frequency of meeting acquaintances** | 0.03 | (-0.90–0.77) | 0.879 | 40.28 | 12,067 | 1,118 | 983 |

CI, confidence interval. The outcome variables of frequency of going out and frequency of meeting acquaintances are continuous, ranging from 1 (not at all) to 6 (almost every day). Other outcome variables are binary, with one for conducting the behavior and zero otherwise.

associated with protective behaviors in the first round of the survey, which is consistent with our expectations (S6 Table).

## Discussion

This study investigated the influence of vaccination on infection-preventive behaviors against COVID-19 by applying a fuzzy RDD; we utilized Japan's vaccination procedure in which those born before April 1, 1957, were prioritized to receive vaccination. The results showed no significant changes in participants' infection-preventive behaviors after vaccination, including wearing facemasks, handwashing, and avoiding going out, in July 2021.

This study offers two significant contributions to the existing literature. First, we examined the causal effects of vaccination on implementing preventive behaviors against COVID-19 by applying a fuzzy RDD. Almost no studies have investigated the relationship between vaccination and preventive behaviors using quasi-experimental designs such as RDD. One exception is a working paper that used RDD for the Japanese population [31]. However, its main focus was on the people's support for the government. Their results were similar to ours as they did not observe an effect of vaccination on infection-protective behaviors.

Second, this study used 14 indicators to measure preventive behaviors. By broadening our focus beyond commonly observed behaviors, we gained a more comprehensive understanding of the impact of vaccination on preventive behaviors in Japan.

Although this study did not identify any evidence of risk-compensation behaviors [5] caused by COVID-19 vaccination, the findings need to be interpreted within the context of the specific situation in Japan during the survey period. The fourth round of the RIETI questionnaire survey, on which this study mainly depended, was conducted from July 20 to 27, 2021. During this period, it is likely that a significant number of Japanese individuals experienced a heightened sense of urgency due to COVID-19, possibly contributing to the ongoing practice of protective behaviors against the virus. Moreover, a state of emergency was announced in multiple areas of Japan, including Tokyo on July 12, 2021, and Okinawa on May 23, 2021.

Priority measures to prevent disease spread were implemented in many other areas. In August 2021, Japan experienced substantial excess mortality for the first time since the COVID-19 outbreak, which may have been caused by the spread of the Delta strain [32]. Therefore, it may be difficult for Japan's situation in July 2021 to be generalized to other countries and times.

This study is subject to several limitations. First, the findings are limited to those aged around 65 years, owing to the innate limitations of the RDD. In addition, the results are limited to "compliers" who would not be treated if their running value were under the cutoff and would be treated if their value were beyond the cutoff. Second, there may be inherent differences between being born before April 1, 1957, and after that date. For example, the school enrollment period in Japan is from April 2 of a year to April 1 of the following year. Therefore, those born before April 1 attend elementary school one year earlier than those born just after April 2. We cannot completely rule out the possibility that this difference may have affected the results of this study. Third, the participants may not be fully representative of the general population. The study primarily relies on data from the fourth round of the RIETI questionnaire survey with participation limited to individuals who responded in both the first and fourth survey rounds. Moreover, this study was conducted online, requiring participants to have Internet access. Fourth, we did not have data regarding when the study participants received the vaccination. The vaccination effectsbegin ten days or more after the vaccination [32]. Finally, as mentioned earlier, in August 2021, Japan experienced substantial excess mortality for the first time since the COVID-19 outbreak, which may have been caused by the spread of the Delta strain [33]. Therefore, it may be difficult for Japan's situation in July 2021 to be generalized to other countries and times.

Therefore, infection-preventive behaviors may differ depending on the number of days since vaccination. However, this possibility could not be accommodated in this study.

## Conclusions

This study found no evidence of risk-compensation behaviors caused by vaccination against COVID-19 using a quasi-experimental design (fuzzy RDD). However, careful consideration is required to ensure the study's generalizability.

## Supporting information

**S1 Fig. Distribution of birth months.**
(TIF)

**S1 Table. Definitions of variables.**
(DOCX)

**S2 Table. Characteristics of study participants in the first survey round.**
(DOCX)

**S3 Table. Estimation result of the effect of eligibility on vaccination rates (treatment: Twice).**
(DOCX)

**S4 Table. Estimation results of the effect of eligibility on vaccination rates (treatment: At least once).**
(DOCX)

**S5 Table. Estimation results of regression discontinuity design on outcomes (treatment: At least once).**
(DOCX)

**S6 Table. Estimation results of regression discontinuity design on outcomes (the first survey round).**
(DOCX)

# Acknowledgments

We would like to thank the participants of this study.

# Author Contributions

**Conceptualization:** Fengming Chen, Hayato Nakanishi, Yoichi Sekizawa.

**Data curation:** Yoichi Sekizawa, Sae Ochi, Mirai So.

**Formal analysis:** Fengming Chen, Hayato Nakanishi, Yoichi Sekizawa.

**Funding acquisition:** Fengming Chen.

**Investigation:** Fengming Chen, Hayato Nakanishi, Yoichi Sekizawa.

**Methodology:** Fengming Chen, Hayato Nakanishi, Yoichi Sekizawa.

**Project administration:** Fengming Chen.

**Resources:** Sae Ochi, Mirai So.

**Software:** Fengming Chen, Yoichi Sekizawa.

**Supervision:** Sae Ochi, Mirai So.

**Validation:** Fengming Chen, Yoichi Sekizawa.

**Visualization:** Fengming Chen, Yoichi Sekizawa.

**Writing – original draft:** Fengming Chen, Hayato Nakanishi, Yoichi Sekizawa.

**Writing – review & editing:** Fengming Chen, Hayato Nakanishi, Yoichi Sekizawa, Sae Ochi, Mirai So.

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
