## [Decision Letter · Decision Letter 0]

31 Jan 2024

PONE-D-23-30545Investigating the causal effects of COVID-19 vaccination on the adoption of protective behaviors in Japan: Insights from a fuzzy regression discontinuity designPLOS ONE

Dear Dr. Chen,

Thank you for submitting your manuscript to PLOS ONE. After careful consideration, we feel that it has merit but does not fully meet PLOS ONE’s publication criteria as it currently stands. Therefore, we invite you to submit a revised version of the manuscript that addresses the points raised during the review process.

The paper investigates the impact of COVID-19 vaccination on how individuals behave using a Regression Discontinuity Design (RDD). The reviewers make several indications to improve the paper, focusing on specific issues, how easy it is to access the data, and how others can replicate the study. The abstract needs a complete redo. It should be clearer, highlight the most important findings, and explain why these findings matter. This means ensuring the main results stand out and showing how they add to our knowledge. One of the reviewers has expressed a deep concern over data and code availability, emphasizing the importance of transparency. The reviewer highlights the importance of replicability in scientific production. The reviewer stresses that being open and able to replicate studies is crucial to benefit the scientific community.

We look forward to receiving your revised manuscript.

Kind regards,

Ivan Filipe de Almeida Lopes Fernandes, Ph.D.

Academic Editor

PLOS ONE

Journal Requirements:

Additional Editor Comments:

The paper investigates the impact of COVID-19 vaccination on how individuals behave using a Regression Discontinuity Design (RDD). The reviewers make several indications to improve the paper, focusing on specific issues, how easy it is to access the data, and how others can replicate the study. The abstract needs a complete redo. It should be clearer, highlight the most important findings, and explain why these findings matter. This means ensuring the main results stand out and showing how they add to our knowledge. One of the reviewers has expressed a deep concern over data and code availability, emphasizing the importance of transparency. The reviewer highlights the importance of replicability in scientific production. The reviewer stresses that being open and able to replicate studies is crucial to benefit the scientific community.

Reviewers' comments:

Reviewer's Responses to Questions

**Comments to the Author**

1. Is the manuscript technically sound, and do the data support the conclusions?

Reviewer #1: Yes

Reviewer #2: No

2. Has the statistical analysis been performed appropriately and rigorously? 

Reviewer #1: Yes

Reviewer #2: I Don't Know

3. Have the authors made all data underlying the findings in their manuscript fully available?

Reviewer #1: Yes

Reviewer #2: No

4. Is the manuscript presented in an intelligible fashion and written in standard English?

Reviewer #1: Yes

Reviewer #2: Yes

5. Review Comments to the Author

Reviewer #1: The manuscript seeks to understand the relationship between vaccination and protective behaviors against Covid-19. It uses an interesting, little-used methodology that deserves to be highlighted. Few changes are suggested in the text.

Pag. 3, line 53 (Introduction):Correct this sentence. Vaccination does not prevent infection in anyone, it only improves the individual's immune response

Reviewer #2: The paper utilizes RDD (Regression Discontinuity Design) to estimate the impact of COVID-19 vaccination on the protective behaviors of individuals in Japan. In this review, I aim to enhance the article's quality by addressing some critical issues.

Regarding the abstract, I suggest the following improvements:

a) Clarity and Brevity: simplify the abstract by focusing on the primary findings and the study's significance. This could involve shortening sentences and eliminating non-essential details for improved clarity and readability. For example, there's no need to repeat "quasi-experimental design" twice in the same section (background).

b) Emphasize Key Results: clearly articulate the main study findings in the abstract, especially those related to the influence of COVID-19 vaccination on protective behaviors. This will provide readers with a quick grasp of the study's outcomes. The authors have adequately reported the main results, but I believe it would be more informative to reference other studies on the topic and explain how the results confirm or contradict existing evidence on the subject.

c) Implications and Context: briefly touch upon the broader implications of the findings and their contributions to existing knowledge or public health policies. This will help readers understand the research's significance within the broader context of managing the COVID-19 pandemic.

Regarding data accessibility, I am interested in whether the data and code are publicly available. Transparency is a fundamental aspect of scientific research, and there is no reason to keep this data confidential. It's crucial to emphasize that my acceptance of the paper is contingent on the availability of data and code in a public repository, such as GitHub or Dataverse.

Upon further examination of the submission, I came across the statement: "Data cannot be publicly shared because it is owned by the RIETI, and the authors lack permission to do so. The data can be accessed through the RIETI by researchers who meet the criteria for accessing confidential data."

Unfortunately, due to this data limitation, I must recommend rejecting the paper. PLOS ONE is a highly prestigious journal, and I believe that a focus on open data policies will further enhance its prestige. Without accessible data, it becomes impossible to verify the reliability of the results. If we cannot scrutinize the data, we cannot challenge or validate your findings, which goes against the principles of scientific inquiry.

Nobody likes to have a paper rejected. I understand that. However, the entire scientific community will benefit if scientific production increasingly adheres to strict rules of transparency and replicability.

Best regards,

6. PLOS authors have the option to publish the peer review history of their article (what does this mean?). If published, this will include your full peer review and any attached files.

Reviewer #1: No

Reviewer #2: No

---

## [Author Response · Author response to Decision Letter 0]

11 Mar 2024

We would like to express our sincere gratitude to both the reviewers and the editor for their extremely helpful and constructive comments on our manuscript, which have helped improve it significantly. We have done our best to address every point raised by the reviewers.

Reviewer #1: 

The manuscript seeks to understand the relationship between vaccination and protective behaviors against Covid-19. It uses an interesting, little-used methodology that deserves to be highlighted. Few changes are suggested in the text.

(Comment 1) Pag. 3, line 53 (Introduction): Correct this sentence. Vaccination does not prevent infection in anyone, it only improves the individual's immune response.

Response 1: Thank you for the reviewer’s comment. We have revised this inaccurate statement and replaced the reference as follows. 

Pag.4, lines 73-80(Manuscript_change track):

”While vaccination decreases the likelihood of hospitalization and may mitigate COVID-19 symptoms, it is not an infallible solution to the issue of COVID-19. This is because, following vaccination, some individuals continue to exhibit weak immune responses[1]. Hence, there is a concern that people who believe a lower risk of being infected or experiencing complications through vaccination take fewer protective measures, which may paradoxically lead to the spread of the virus [2–3].”

1.Lipsitch M, Krammer F, Regev-Yochay G, Lustig Y, Balicer RD. SARS-CoV-2 breakthrough infections in vaccinated individuals: Measurement, causes and impact. Nat Rev Immunol. 2022;22:57–65. doi:10.1038/s41577-021-00662-4

Response 2: We conducted a thorough review of the reference list and main text, implementing minor revisions focused mainly on the inclusion of DOIs and the correction of inaccurate statements. Previously omitted, DOIs have now been incorporated into the revised manuscript.

Reviewer #2:

The paper utilizes RDD (Regression Discontinuity Design) to estimate the impact of COVID-19 vaccination on the protective behaviors of individuals in Japan. In this review, I aim to enhance the article's quality by addressing some critical issues.

(Comment 1)Regarding the abstract, I suggest the following improvements:

a) Clarity and Brevity: simplify the abstract by focusing on the primary findings and the study's significance. This could involve shortening sentences and eliminating non-essential details for improved clarity and readability. For example, there's no need to repeat "quasi-experimental design" twice in the same section (background).

b) Emphasize Key Results: clearly articulate the main study findings in the abstract, especially those related to the influence of COVID-19 vaccination on protective behaviors. This will provide readers with a quick grasp of the study's outcomes. The authors have adequately reported the main results, but I believe it would be more informative to reference other studies on the topic and explain how the results confirm or contradict existing evidence on the subject.

c) Implications and Context: briefly touch upon the broader implications of the findings and their contributions to existing knowledge or public health policies. This will help readers understand the research's significance within the broader context of managing the COVID-19 pandemic. 

Response 1: Thank you for the reviewer’s extensive suggestions on the abstract. Following the advice, we rewrote the abstract as follows.

Pag.2-3, lines, 21-63(Manuscript_change track):

“Abstract

Background: During the COVID-19 pandemic, concerns emerged that vaccinated individuals might engage less in infection-preventive behaviors, potentially contributing to virus transmission. This study evaluates the causal effects of COVID-19 vaccination on such behaviors within Japan, highlighting the significance of understanding behavioral dynamics in public health strategies.

Methods: Utilizing Japan's age-based vaccination priority for those born before April 1, 1957, this research employs a regression discontinuity design (RDD) to assess the vaccination's impact. Data from the fourth round of a longitudinal online survey, conducted by the Research Institute of Economy, Trade, and Industry from July 20 to 27, 2021, served as the basis for analyzing 14 infection protective behaviors, including mask usage, handwashing, and avoiding crowds.

Results: A total of 12067 participants completed the survey. The analyzed sample size varied by outcome variable, ranging from 1499 to 5233. Analysis revealed no significant differences in the 14 behaviors examined among fully vaccinated, partially vaccinated, and unvaccinated individuals. This consistency across groups suggests that vaccination status did not significantly alter engagement in protective behaviors during the observation period.

Conclusions: Empirical findings highlight the complexity of behavioral responses following vaccination, indicating that such responses may be influenced by various factors, rather than vaccination status alone. Additionally, this result underscores the importance of crafting public health policies that account for the intricate interplay between vaccination and behavior. This study contributes to the broader discourse on managing responses to the pandemic and tailoring interventions to sustain or enhance protective health behaviors amid vaccination rollouts.”

(Comment 2)Regarding data accessibility, I am interested in whether the data and code are publicly available. Transparency is a fundamental aspect of scientific research, and there is no reason to keep this data confidential. It's crucial to emphasize that my acceptance of the paper is contingent on the availability of data and code in a public repository, such as GitHub or Dataverse.

Upon further examination of the submission, I came across the statement: "Data cannot be publicly shared because it is owned by the RIETI, and the authors lack permission to do so. The data can be accessed through the RIETI (contact via e-mail: info@rieti.go.jp)by researchers who meet the criteria for accessing confidential data."

Unfortunately, due to this data limitation, I must recommend rejecting the paper. PLOS ONE is a highly prestigious journal, and I believe that a focus on open data policies will further enhance its prestige. Without accessible data, it becomes impossible to verify the reliability of the results. If we cannot scrutinize the data, we cannot challenge or validate your findings, which goes against the principles of scientific inquiry.

Response 2: As a response to the reviewer’s concern regarding data availability, which is a critical aspect of the peer review process and necessary for replication studies, we acknowledge the importance of sharing our data. 

The dataset in question is owned by the Research Institute of Economy, Trade, and Industry (RIETI). We have consulted with RIETI officers responsible for data management, and they have not approved the disclosure of the data for the following reasons:

1. RIETI does not have the policy and guidelines to make the data public.

2. Although the dataset is anonymous, certain combinations of variables, such as birth year and month, and postal code of residence, may lead to the identification of personal information. Revealing personal information from the dataset would violate the agreement with the study participants and the Japanese Personal Information Protection Law.

Therefore, RIETI officers must be cautious about making the data public. However, while the data are not freely available to everyone, researchers who meet the criteria for accessing confidential data can obtain access through RIETI. This requires approval from the ethical committee of the hospital, a commitment to strict data management for confidentiality protection, and other procedural commitments. We are prepared to assist researchers interested in accessing the data from RIETI. 

Additionally, we have made the Stata codes available via protocols.io as follows:doi:dx.doi.org/10.17504/protocols.io.yxmvm38pnl3p/v1

Response 3: We conducted a thorough review of the reference list and main text, implementing minor revisions focused mainly on the inclusion of DOIs and the correction of inaccurate statements. Previously omitted, DOIs have now been incorporated into the revised manuscript.

---

## [Editor Report · Decision Letter 1]

27 Mar 2024

PONE-D-23-30545R1Investigating the causal effects of COVID-19 vaccination on the adoption of protective behaviors in Japan: Insights from a fuzzy regression discontinuity designPLOS ONE

Dear Dr. Chen,

Thank you for submitting your manuscript to PLOS ONE. After careful consideration, we feel that it has merit but does not fully meet PLOS ONE’s publication criteria as it currently stands. Therefore, we invite you to submit a revised version of the manuscript that addresses the points raised during the review process. 

I still understand that there are questions about the replicability of the study proposed here. I would ask you to incorporate into the text of the manuscript information relevant to obtaining the data, following the recommendations of the Plos One Data Availability Policy (https://journals.plos.org/plosone/s/data-availability):

I would also be interested in some clarification on the sentence "We requested that all eligible respondents from the first round of the survey participate in the 132 subsequent survey rounds via email" on page 06 of the revised version of the manuscript. The sentence indicates that the authors were part of the data collection process. If this was the case, it is no longer a third part dataset, as was first understood.

We look forward to receiving your revised manuscript.

Kind regards,

Ivan Filipe de Almeida Lopes Fernandes, Ph.D.

Academic Editor

PLOS ONE

Journal Requirements:

Additional Editor Comments:

Dear authors,

Thank you very much for your efforts to address the points made by the two reviewers. Nonetheless, I understand that there are still questions about the replicability of the proposed study.

I would ask you to incorporate into the text of the manuscript information relevant to obtaining the data, following the recommendations of the Plos One Data Availability Policy (https://journals.plos.org/plosone/s/data-availability):

I ask for special attention to the section: Third-party data

For studies involving third-party data, we encourage authors to share any data specific to their analyses that they can legally distribute. PLOS recognizes, however, that authors may be using third-party data they do not have the rights to share. When third-party data cannot be publicly shared, authors must provide all information necessary for interested researchers to apply to gain access to the data.

3) All necessary contact information others would need to apply to gain access to the data

Authors should properly cite and acknowledge the data source in the manuscript. Please note, if data have been obtained from a third-party source, we require that other researchers would be able to access the data set in the same manner as the authors.

I would also be interested in some clarification on the sentence "We requested that all eligible respondents from the first round of the survey participate in the 132 subsequent survey rounds via email" on page 06 of the revised version of the manuscript.

The sentence indicates that the authors were part of the data collection process. If this was the case, it is no longer a third part dataset, as was first understood.

---

## [Author Response · Author response to Decision Letter 1]

4 May 2024

We would like to express our sincere gratitude to the editor for their insightful and constructive feedback on our manuscript. Following the suggestions, we have carefully revised the manuscript and addressed each comment in detail. We provide a point-by-point response to issues raised as follows. 

(Comment 1) I still understand that there are questions about the replicability of the study proposed here. I would ask you to incorporate into the text of the manuscript information relevant to obtaining the data, following the recommendations of the Plos One Data Availability Policy.

Response 1: We appreciate this comment and have updated the manuscript accordingly. Specifically, on page 7, lines 145–154 (Manuscript_change track), we have included detailed information about how the dataset, managed by RIETI, is not publicly accessible due to legal and policy constraints. However, RIETI permits data access to researchers who meet strict confidentiality criteria. Full details and contact information for data access requests are now clearly stated in the manuscript.

(Comment 2) I would also be interested in some clarification on the sentence "We requested that all eligible respondents from the first round of the survey participate in the subsequent survey rounds via email" on page 06 of the revised version of the manuscript. The sentence indicates that the authors were part of the data collection process. If this was the case, it is no longer a third part dataset, as was first understood.

Response 2: Thank you for highlighting this need for clarity. To address this, we have revised the manuscript pages 6–7, lines 117–138 (Manuscript_change track) to clarify that although YS, a co-author, was involved in designing the survey and managing the data collection process as an employee of RIETI, the actual data ownership and rights to distribute remain with RIETI. This makes the dataset “third-party data” under PLOS ONE’s definitions, as the authors do not own the dataset nor have the rights to freely distribute it. Detailed explanations of the author roles and processes have been added to ensure transparency.

Editorial Request: Please ensure that your Response to Reviewers letter includes a point-by-point response to all comments provided by the editor and reviewers.

Response 3: We have reviewed the entire manuscript and reference list, making the necessary revisions including the addition of two new references and corrections to inaccuracies pointed out during the review. Each point raised by the reviewers has been addressed as shown in this letter, and we believe the revisions have greatly improved the manuscript.

---

## [Editor Report · Decision Letter 2]

23 May 2024

Investigating the causal effects of COVID-19 vaccination on the adoption of protective behaviors in Japan: Insights from a fuzzy regression discontinuity design

PONE-D-23-30545R2

Dear Dr. Chen,

We’re pleased to inform you that your manuscript has been judged scientifically suitable for publication and will be formally accepted for publication once it meets all outstanding technical requirements.

Kind regards,

Ivan Filipe de Almeida Lopes Fernandes, Ph.D.

Academic Editor

PLOS ONE
---

## [Editor Report · Acceptance letter]

3 Jun 2024

PONE-D-23-30545R2 

PLOS ONE

Dear Dr. Chen, 

I'm pleased to inform you that your manuscript has been deemed suitable for publication in PLOS ONE. Congratulations! Your manuscript is now being handed over to our production team.

Kind regards, 

on behalf of

Dr. Ivan Filipe de Almeida Lopes Fernandes 

Academic Editor

PLOS ONE